# Variation in Alanine Aminotransferase in Children with Non-Alcoholic Fatty Liver Disease

**DOI:** 10.3390/children9030374

**Published:** 2022-03-08

**Authors:** Eduardo Castillo-Leon, Heather L. Morris, Cheryl Schoen, Jacob Bilhartz, Patrick McKiernan, Tamir Miloh, Sirish Palle, Mohammad Nasser Kabbany, Breda Munoz, Andrea R. Mospan, Bryan Rudolph, Stavra A. Xanthakos, Miriam B. Vos

**Affiliations:** 1Department of Pediatrics, School of Medicine, Emory University, Atlanta, GA 30322, USA; eduardo.castillo.leon@emory.edu; 2Department of Pediatrics, Morehouse School of Medicine, Atlanta, GA 30310, USA; 3Target RWE, Durham, NC 27703, USA; hmorris@targetrwe.com (H.L.M.); cschoen@targetrwe.com (C.S.); bmunozhernandez@targetrwe.com (B.M.); amospan@targetrwe.com (A.R.M.); 4Department of Pediatrics, University of Michigan, Ann Arbor, MI 48109, USA; jacobbil@med.umich.edu; 5Children’s Hospital of Pittsburgh, Pittsburgh, PA 15224, USA; patrick.mckiernan@chp.edu; 6Pediatric Gastroenterology, Pediatric Transplant Hepatology, Miami Transplant Institute, Miami, FL 33136, USA; txm760@med.miami.edu; 7Division of Gastroenterology, OU Medicine, Oklahoma City, OK 73104, USA; sirish-palle@ouhsc.edu; 8Department of Pediatric Gastroenterology, Hepatology, and Nutrition, Cleveland Clinic, Cleveland, OH 44195, USA; kabbanm@ccf.org; 9The Children’s Hospital at Montefiore, The Pediatric Hospital for Albert Einstein College of Medicine, Bronx, NY 10467, USA; brudolph@montefiore.org; 10Division of Gastroenterology, Hepatology and Nutrition, Cincinnati Children’s, Department of Pediatrics, College of Medicine, University of Cincinnati, Cincinnati, OH 45229, USA; stavra.xanthakos@cchmc.org; 11Nutrition and Health Sciences Program, Laney Graduate School, Emory University, Atlanta, GA 30322, USA

**Keywords:** cirrhosis, non-alcoholic fatty liver disease, ALT

## Abstract

Background: Pediatric non-alcoholic fatty liver disease (NAFLD) is a major public health concern. Aminotransferase (ALT) is frequently used for screening and monitoring, but few studies have reported typical patterns of ALT elevation in children. Methods: TARGET-NASH is a real-world longitudinal observational cohort of patients with NAFLD receiving care across the United States. Analyses included children enrolled between 1 August 2016, and 12 October 2020, with at least one ALT measurement after enrollment. Peak ALT was based on the first and last available record and categorized into clinical cut points: <70 IU/L, >70–<250 IU/L, and >250 IU/L. A chi-squared test was used to compare differences in proportions, and a Kruskal–Wallis test was used to compare the medians and distributions of continuous responses. Results: Analyses included 660 children with a median age of 13 years. Of the 660, a total of 187 had undergone a biopsy and were more likely to be Hispanic or Latino (67% vs. 57%, *p* = 0.02) and to have cirrhosis (10% vs. 1%, *p* < 0.001). The highest ALT scores ranged from 28 U/L to 929 U/L; however, these scores varied across time. The prevalence of cirrhosis or any liver fibrosis stage was most common among children with a peak ALT > 70 U/L. Conclusions: Large variability was seen in ALT among children, including many values > 250 U/L. Higher levels of ALT were associated with increased prevalence of comorbidities and more advanced stages of NAFLD. These findings support an increased need for therapeutics and disease severity assessment in children with peak ALT > 70 U/L.

## 1. Introduction

Pediatric non-alcoholic fatty liver disease (NAFLD) is a major public health problem that has raised concern worldwide. In recent years, NAFLD has developed into one of the most common chronic diseases affecting children as young as two years of age, and the prevalence is continuing to increase [1,2,3]. The rate of pediatric patients with NAFLD has more than doubled in recent decades and has persisted across gender and racial/ethnic subgroups [4]. A meta-analysis of studies examining the prevalence of NAFLD worldwide showed a pooled prevalence of 7.6% in the general population and approximately 36% among children with obesity. With the increased prevalence of obesity among pediatric patients worldwide, the rate of NAFLD could continue to rise in the coming years [1,5,6].

While the risk factors associated with NAFLD in pediatric patients are similar to adults, the presence of either metabolic syndrome (MetS) or central adiposity places children at an increased risk for severe steatosis, inflammation, and advanced fibrosis [7,8]. In children, the long-term implications and trajectory of the disease are still unknown; however, some studies suggest that NAFLD presenting in childhood may be more severe compared to disease that initiates in adulthood [9]. For example, at diagnosis, 15% of the children have advanced liver fibrosis compared to 11.5% of adults [10,11], a crucial histologic feature that predicts mortality in adult patients with NAFLD.

Alanine aminotransferase (ALT) is commonly used both to screen and monitor the severity of NAFLD in clinical practice and clinical trials due in part to its availability and cost-effectiveness [10]. Despite the common utilization of ALT, there is a paucity of data describing both the range and clinical characteristics associated with ALT elevation among children with NAFLD in “real-world” practice. While the median or mean ALT for a cohort is commonly reported, it is less well documented what percentage of children with NAFLD have very high ALT levels. Further, ALT is commonly used as an inclusion criterion in NAFLD clinical trials, and ALT > 250 U/L is frequently an exclusion criterion due to insufficient data on children with levels > 250 U/L. This study describes the distribution of peak ALT levels as well as the distribution of clinical characteristics by both history of a biopsy and peak serum ALT category (≤70, >70 to ≤250, and >250 U/L) among children enrolled in the real-world data cohort TARGET-NASH.

## 2. Methods

### 2.1. Cohort

TARGET-NASH is a real-world longitudinal observational cohort of pediatric and adult patients with NAFLD receiving care in both academic and community Hepatology, Gastroenterology, and Endocrinology practices across the United States. Upon enrollment, patients who consent to participate provide access to their medical records for three years prior to the date of enrollment and are then followed prospectively for up to five years. Patient narratives, laboratory values, pathology reports, and imaging data are extracted from the medical record and uploaded into a secured database at six-month intervals from the date of enrollment. Demographics, comorbidities, concomitant medications, interventions for NAFLD, and liver disease progression are represented in the database as well as adverse outcomes, including cardiovascular and neoplastic complications. A detailed description of the TARGET-NASH study design has been previously published [12,13]. Approvals from central and/or local institutional review boards were obtained prior to subject recruitment and enrollment.

This analysis included children enrolled in TARGET-NASH between 1 August 2016 and 12 October 2020 receiving care across 12 academic specialty hepatology and gastroenterology pediatric institutions in the US with at least one ALT measurement from the date of enrollment. For peak ALT and follow-up derivations, any ALTs collected more than 3.5 years prior to the date of enrollment are excluded.

### 2.2. Liver Disease Case Definition

All patients had a diagnosis of NAFLD at enrollment by their treating physician based on either liver biopsy, imaging, or a pragmatic case definition and were being managed in usual clinical practice. Keeping in line with the real-world data approach, the NAFLD definition used reflects what occurs in common practice rather than within guideline recommendations. A categorization of cirrhosis was based on findings from liver biopsies or imaging with a clinical designation of cirrhosis.

### 2.3. Patient Characteristics

Demographics including age, gender, ethnicity, any medication use, as well as any history of comorbid disease were obtained at the time of enrollment. Body mass index (BMI) z-score and weight percentile were calculated based on the most recent height and weight values closest to enrollment using the Centers for Disease Control 2000 growth charts. Weight category was derived from BMI percentiles, with classification as underweight (BMI < 5 percentile), normal weight (BMI 5 to <85 percentile), overweight (BMI 85 to <95 percentile), and obese (BMI ≥ 95 percentile).

Laboratory measures including aspartate aminotransferase (AST), ALT, albumin, total bilirubin, alkaline phosphatase (ALP), hemoglobin, and platelet count were ascertained from the values closest to enrollment. Liver histology was obtained from the most recent radiographic and biopsy reports.

### 2.4. Alanine Aminotransferase (ALT)

Peak ALT was ascertained over the longitudinal follow-up period from the child’s first medical record on file to the last available record on file. ALT was categorized into a three-level variable according to the following a priori clinical cut points: ≤70 IU/L, >70–≤ 250 IU/L, and >250 IU/L. These cutoffs are based on inclusion criteria commonly used in clinical trials of children with NAFLD. These inclusion criteria usually require an ALT value higher than three times the upper limit of normal and lower than ten times the upper limit normal. For patients with a minimum of five ALT measurements at least 1 month apart, ALT trajectories across each of the clinical cut points (≤70 IU/L, >70–≤250 IU/L, >250 IU/L) were generated. ALTs collected within 3 months of the onset of cholestasis, cholelithiasis, pancreatitis, sepsis, bariatric surgery, and other conditions indicative of gallbladder disease were excluded from the ALT trajectory analysis as the ALT level may be indicative of these conditions and not the underlying liver disease. A secondary review was performed on records obtained from three patients with the highest ALT values (929, 819, 715). All three patients had clinical data and liver biopsies performed indicating NASH without evidence of other etiologies and were therefore included in the study.

### 2.5. Statistical Analysis

The mean and standard deviations were calculated across continuous measures of patient characteristics, while counts and percentages were provided for categorical responses.

A chi-squared test was used to compare the differences in proportions across categories of patient characteristics. For cell counts less than 5, a Fisher’s exact test was used. A Kruskal–Wallis test was used to compare the medians and distributions of continuous responses, and a Fisher’s exact test was used for group comparisons. A *p*-value of <0.05 was considered statistically significant. All analyses were performed using SAS version 9.4 (SAS Institute, Cary, NC, USA).

## 3. Results

### 3.1. Study Population

The analysis included 660 children diagnosed with NAFLD and enrolled in TARGET-NASH between 1 August 2016 and 12 October 2020 who had at least one ALT measurement. The median age of children in the cohort was 13 years, and 67% were white, 60% were Hispanic or Latino, and 70% were male. The median BMI z-score was 2.36 (range 0–4.5), and 93% of children were obese. The prevalence of NAFLD cirrhosis was 3.6%, and 18.6% of children were diagnosed with Type 2 diabetes (Table 1). Of the 660 patients, 101 patients (15.3%) had a history of anxiety or depression, 89 patients (13.5%) had a history of pediatric ADHD, and 71 patients (10.8%) had a history of psoriasis or other autoimmune skin diseases.

### 3.2. Patient Characteristics by Receipt of Biopsy

Of the 660 children with NAFLD, a total of 187 had a biopsy reported. Patients who had or had not received a biopsy did not differ significantly regarding age, gender, and race. However, patients who received a biopsy were more likely to be Hispanic or Latino (67% vs. 57%, *p* = 0.02) or have cirrhosis (10% vs. 1%, *p* < 0.001) and diabetes (27% vs. 15%, *p* < 0.001). Children with a reported biopsy were found to have a higher peak ALT and AST than those without a biopsy (ALT: 169.0 U/L vs. 88.0 U/L, *p* < 0.001; AST: 91.0 U/L vs. 52.0 U/L, *p* < 0.001). Laboratory values for albumin, hemoglobin, and ALP were also found to be significantly different between those who did or did not have a biopsy (Table 1).

Biopsy recipients used either metformin (26% vs. 13%, *p* < 0.001) or vitamin E (36% vs. 6%, *p* < 0.001) more frequently and had a longer follow-up time between the first ALT measurement and the last ALT measurement available (median = 34.9 months vs. 26.2 months, *p* < 0.001) (Table 1).

### 3.3. Patient Characteristics by Peak ALT Category

The prevalence of cirrhosis was more common among children with a peak ALT > 70 U/L compared to children with a peak ALT ≤ 70 U/L. Children with a peak ALT between 71 and 250 U/L were 2.8 times as likely to have cirrhosis (compared to ALT ≤ 70 U/L), and children with a peak ALT > 250 U/L were eight times more likely to have cirrhosis compared to children with a peak ALT ≤ 70 U/L. The prevalence of Type 2 diabetes was 2.2 times as likely among children with an ALT > 250 U/L compared to children with a peak ALT between 71 and 250 and children with a peak ALT ≤ 70 U/L (ALT ≤ 70 U/L, 17%; ALT between 71–250, 16%; ALT > 250 U/L, 36%; *p* < 0.001) (Table 2).

The distribution of age, race, ethnicity, and BMI z-score were similar between peak ALT categories; however, the male-to-female ratio was significantly higher among children with a peak ALT > 70 U/L. Among children with an ALT value > 70 U/L, children were nearly twice as likely to be of Hispanic or Latino ethnicity (Hispanic/Latino: 60%, Non-Hispanic: 40%). Median AST and ALP increased across peak ALT categories, and there was a significant difference across peak ALT categories in median hemoglobin and platelets (*p* = 0.002 and *p* = 0.008, respectively). Other than AST, hemoglobin, and platelets, the distribution of biochemical values were not significantly different by peak ALT level. The proportion of children that were prescribed either metformin or vitamin E was higher among children with a peak ALT > 250 U/L than among children with a peak ALT between 71–250 U/L or a peak ALT ≤ 70 U/L (metformin: *p* = 0.001 and vitamin E: <0.001, respectively) (Table 2). The median time (in months) from the first ALT to the last ALT was greatest for patients with a peak ALT > 250 (ALT ≤ 70 U/L, 28.0; ALT > 70–≤250 U/L, 28.3; ALT > 250 U/L, 31.1; *p* = 0.152).

### 3.4. Liver Histology

Of the 187 children who had a liver biopsy, reports obtained from each institution showed that the prevalence of grade 3 steatosis was more than 3 times as likely among children with a peak ALT > 70 U/L compared to children with peak ALT ≤70 U/L (ALT ≤ 70 U/L, 17%; ALT 70–<250 U/L, 51%; ALT ≥ 250 U/L, 50%; *p* = 0.113). The proportion of children with evidence of liver fibrosis stage, assessed by Brunt or NAS scoring, was greater among children in the highest peak ALT category compared to the lowest peak category (ALT ≤ 70 U/L, 58%; ALT > 70–≤250 U/L, 66%; ALT > 250 U/L, 75%; *p* = 0.372). Lobular inflammation grade 2 (>4 foci under 20× field) and hepatocyte ballooning grade 2 (many ballooned cells) were only present among children with ALT >70 U/L. Moreover, the proportion of children with a NAS score of 5 or greater with a peak ALT > 70 U/L (50%) was 4.5 times that of children with a peak ALT < 70 U/L (11%). Of those receiving a biopsy, 85% had steatohepatitis, and 15% had advanced fibrosis (fibrosis stage 3 or greater). Nearly half of all the children who had a biopsy had a NAS total score ≥ 5 (48%) (Appendix A Table A2).

### 3.5. Peak ALT by Patient Characteristics

There were no statistically significant differences in median peak ALT levels between males and females, body mass index percentiles, or age and race categories (Figure 1). Children who identified as being Hispanic or Latino had a peak ALT significantly higher than those who did not (106.0 U/L vs. 98.5 U/L, *p* = 0.018). Peak ALT was also significantly higher in children with any history of metformin (130.0 U/L vs. 100.0 U/L, *p* = 0.003) or Vitamin E use (162.0 U/L vs. 98.0 U/L, *p* < 0.001). The median peak ALT value among children with Type 2 diabetes was higher than children without a history of Type 2 diabetes (137.0 U/L vs. 99.0 U/L, *p* = 0.004). (Figure 1; Table 3).

Children who underwent a liver biopsy (*n* = 187) had a median peak ALT value that was nearly double that of children who did not have a biopsy (*n* = 473) (169.0 vs. 88.0; *p* < 0.001) (Figure 1; Table 1). Additionally, children with steatohepatitis or the presence of fibrosis had a higher median peak ALT value than children without these histological features (Appendix A Table A1).

### 3.6. ALT over Time

Data from pediatric patients with at least five ALT measurements were utilized to investigate the ALT fluctuation across time (Figure 2). Generally, most children who were in the ≤70 IU/L ALT category at their initial measurement (Time 1) stayed at that level over time, with a small percentage moving to the ALT > 250 IU/L category (<1%). Overall, the increasing trend of the percentage of children in the ≤70 IU/L ALT category and the transitions from higher ALT categories to the ≤70 IU/L ALT suggest an improvement in ALT over time. Forty-four percent of children had an ALT between 71 and 250 IU/L at their initial time point and improved slightly over time with some moving into the ALT ≤ 70 IU/L category, while a smaller number of children worsened to an ALT > 250 IU/L at later time points. There was a small number of patients (5%) with ALT > 250 IU/L at their initial time point, and these patients generally remained in the higher ALT category, with only three instances of improving to ≤70 IU/L at some point (Figure 2). Appendix A Table A3 is a numerical representation of the transitions between ALT categories across each of the five time points. The majority of children started with an ALT ≤ 70 IU/L (51%, *n* = 170) followed by 71–250 IU/L (44%, *n* = 148) and >250 IU/L (5%, *n* = 17), and by time point five, these numbers had changed to 54% (*n* = 181), 41% (*n* = 137), and 5% (*n* = 17), respectively (Appendix A Table A3).

## 4. Discussion

This study describes the distribution of clinical and metabolic characteristics among a large cohort of children with NAFLD stratified by receipt of biopsy (biopsy, no biopsy) and peak ALT level (≤70 U/L, >70–≤250 U/L, and >250 U/L). The prevalence of NAFLD, a disease associated with a high risk for premature morbidity and mortality [14], has substantially increased with the increasing rates of obesity in children [1]. Despite the controversial role of ALT in predicting NAFLD severity in children [15,16], ALT is frequently used in clinical practice to monitor disease activity. Overall, there was substantial variability in ALT values among children in this study population. Contrary to trends in adult patients with NAFLD, there were a considerable number of children with NAFLD with ALT values higher than 250 U/L. In this cohort, approximately 12% of patients had a peak ALT > 250 U/L with values ranging between 252 and 929 U/L.

The variations shown in ALT trajectory across categories indicate that the disease state is not static. Patients with higher ALT values were more likely to remain in the higher ALT category and were associated with comorbidities such as Type 2 diabetes and cirrhosis. Conversely, children who initially had ALT < 70 U/L tended to stay in this category and have a less severe disease. While no trends can define the situation for an individual patient, these data suggest a decreased monitoring frequency in children with lower ALT and the importance of further assessment in children with high ALT.

Children who underwent a liver biopsy were more likely to be Hispanic and have higher ALT levels compared to those who did not. This may reflect clinical concern for more significant liver disease driving the decision to perform the biopsy. Hispanic children have the highest risk of NAFLD and appear to have a higher likelihood of advanced liver fibrosis [17,18]. While liver biopsies are the gold standard for assessing disease severity, it is an invasive procedure that has been associated with morbidity, and in some cases, mortality [19,20]. ALT has been shown to have moderate specificity and sensitivity in detecting NAFL, while also being non-invasive and inexpensive [21], with mean ALT values measured over time proving to be a valid biomarker of histologic changes in children [22]. In this study, patients with peak ALT values higher than 70 presented a higher prevalence of cirrhosis, NAS score, and higher grades of fibrosis, inflammation and ballooning. Hispanics were the most prevalent among the highest ALT category. In addition to liver disease severity, the prevalence of Type 2 diabetes was also greater in patients with ALT values higher than 70. Type 2 diabetes has been associated with NAFLD in adult populations, and this can be translated to pediatric populations, as ALT is used as a surrogate for liver fat accumulation. A rise in ALT may reflect not only a greater degree of insulin resistance but also a higher possibility of developing Type 2 diabetes.

The degree of ALT elevation was also associated with a higher prevalence of prescription medication use. The percentage of children that had documented use of metformin, as well as a history of Type 2 diabetes, was higher among children with an elevated ALT (>70 U/L). Vitamin E was more frequently prescribed in children with ALT > 70 U/L. At present, medications are not recommended in the treatment of pediatric patients with NAFLD under the guidance of the NASHPGHAN guidelines [10]. However, like the patients within this study, Vitamin E has been used by clinicians for treating biopsy-proven NASH based on promising results within the adult population [23,24].

This real-world observational study conducted in academic and community sites included a large and diverse multicenter population of children that were diagnosed and managed for NAFLD in usual clinical practice. Previous studies examining pediatric NAFLD have occurred mainly in randomized controlled trials [23,25], thus limiting the sample to a stringent population with very specific inclusion/exclusion criteria and an inability to look at the progression over time in a real-world population [26]. The wide inclusion criteria for this study allowed the analysis of those patients with normal to mildly elevated values of ALT, a population often overlooked.

There were limitations to this study. Not all children in the cohort had a liver biopsy; however, most of them had either an MRI or US of the liver. Both methods are widely used to assess the presence of NAFLD in a clinical setting and in large population-based studies [27,28]. Second, the available biopsies were analyzed by the pathologist in charge of each institution in a routine clinical setting, which might have increased the variability in pathology interpretation. Another limitation was the high percentage of children identified as Hispanics or Latinos, which most likely reflects the distribution of the disease. In this study, even though children of Hispanic or Latino origin are at the highest risk for developing NAFLD [18], the percentage of children who were Hispanic or Latino in each category of peak ALT was similar. This may be due to the diversity of the population of Hispanics or Latinos in this study, misclassification due to the self-reported nature of ethnicity, or the way ALT was categorized using a priori values. Despite these limitations, the findings add to the scientific literature by describing the trends in ALT related to NAFLD during childhood.

In conclusion, pediatric patients with NAFLD enrolled in TARGET-NASH showed large variability in ALT. Peak ALT scores ranged from 10 U/L to 929 U/L, likely influenced both by phase of disease and severity. There were a surprisingly high number of pediatric patients with a peak ALT ≥ 250 within this diverse real-world cohort. Future studies following ALT trajectories over longer time periods and with clinical outcome data would be beneficial for understanding NAFLD among pediatric patients.

## Figures and Tables

**Figure 1 children-09-00374-f001:**
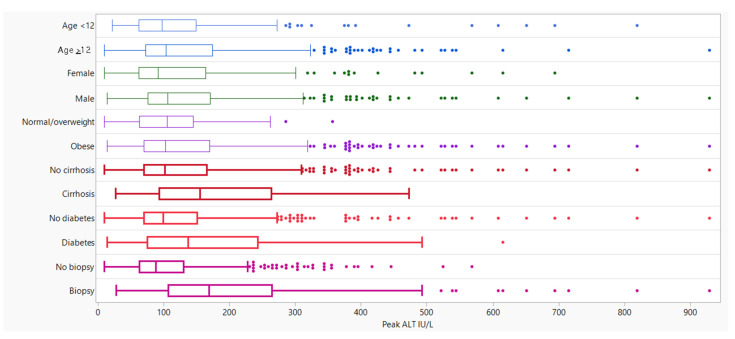
Median and Quartile Range (q1 and q3) of Peak ALT by Patient Characteristics. Box plots with a bolded line indicate the significance differences in mean peak ALT across response values at a level of 0.05 (refer to Table 3).

**Figure 2 children-09-00374-f002:**
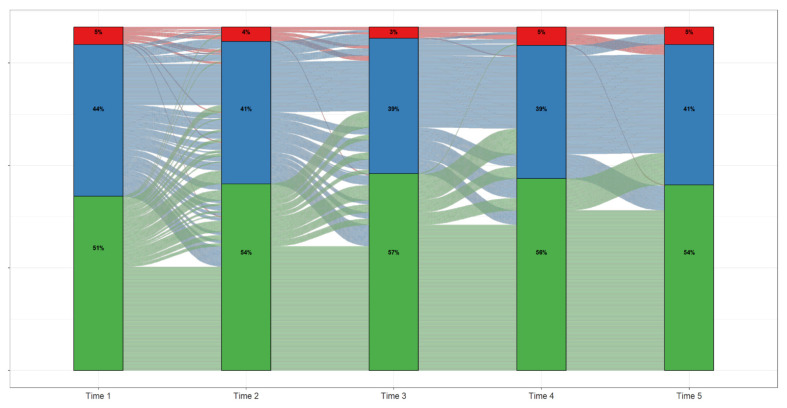
ALT Trajectories Among Pediatric Patients Enrolled in TARGET-NASH.

**Table 1 children-09-00374-t001:** Demographics and Characteristics of Pediatric Patients Enrolled in TARGET-NASH.

Summary	Any Biopsy Reported		All Participants (*n* = 660)	*p*-Values
	Biopsy	No Biopsy		
	(*n* = 187)	(*n* = 473)		
**Clinical Characteristics**				
Age at Study Entry (years) ^†^				0.557
Median (*n*)	14.0 (187)	13.0 (473)	13.0 (660)	
Mean (SD)	13.6 (2.81)	13.4 (3.02)	13.5 (2.96)	
Min–Max	7–21	4–20	4–21	
Age Categories, *n* (%)				0.428
Age < 12	45 (24.1%)	128 (27.1%)	173 (26.2%)	
Age ≥ 12	142 (75.9%)	345 (72.9%)	487 (73.8%)	
Gender, *n* (%)				0.358
Female	61 (32.6%)	137 (29.0%)	198 (30.0%)	
Race, *n* (%)				White vs. Non-White Overall: 0.697
White	122 (65.2%)	319 (67.4%)	441 (66.8%)	
Black or African American	3 (1.6%)	16 (3.4%)	19(2.9%)	
American Indian or Alaska Native	5 (2.7%)	11 (2.3%)	16 (2.4%)	
Asian	5 (2.7%)	5 (1.1%)	10 (1.5%)	
Native Hawaiian or other Pacific Islander	1 (0.5%)	2 (0.4%)	3 (0.5%)	
Other	34 (18.2%)	82 (17.3%)	116 (17.6%)	
Not Reported	17 (9.1%)	38 (8.0%)	55 (8.3%)	
Ethnicity, *n* (%)				0.016
Hispanic or Latino	126 (67.4%)	269 (56.9%)	395 (59.8%)	
Not Hispanic or Latino	61 (32.6%)	201 (42.5%)	262 (39.7%)	
Not Reported	0 (0.0%)	3 (0.6%)	3 (0.5%)	
Insurance Type, *n* (%)	187 (100.0%)	473 (100.0%)	660 (100.0%)	Participants have more than one insurance type so *p*-values are not presented.
Medicaid/Medicare	149 (79.6%)	326 (68.9%)	475 (71.9%)	
Private	36 (19.3%)	133 (28.1%)	169 (25.6%)	
Other	6 (3.2%)	15 (3.1%)	21 (3.2%)	
Uninsured	0 (0.0%)	2 (0.4%)	2 (0.3%)	
Percentile for Most Recent BMI				0.223
Median (*n*)	99.17 (185)	99.05 (467)	99.09 (652)	
Min–Max	87.5–99.9	48.5–100.0	48.5–100.0	
Z-Score for Most Recent BMI				0.158
Median (*n*)	2.40 (185)	2.35 (467)	2.36 (652)	
Min–Max	1.1–3.3	0.0–4.5	0.0–4.5	
Cirrhosis, *n* (%)	18 (9.6%)	6 (1.3%)	24 (3.6%)	<0.001
Diabetes, *n* (%) ^‡^	51 (27.3%)	72 (15.2%)	123 (18.6%)	<0.001
**Laboratory Parameters**				
Lowest Platelets (10^3^/uL)				0.167
Median (*n*)	264.0 (187)	274.0 (452)	272.0 (639)	
Min–Max	12–450	3–571	3–571	
Lowest Albumin (g/dL)				0.016
Median (*n*)	4.1 (187)	4.3 (468)	4.2 (655)	
Min–Max	2–5	1–5	1–5	
Lowest Hemoglobin (g/dL)				0.006
Median (*n*)	13.2 (187)	13.4 (449)	13.3 (636)	
Min–Max	7–17	6–18	6–18	
Highest ALT (U/L)				<0.001
Median (*n*)	169.0 (187)	88.0 (473)	103.0 (660)	
Min–Max	28–929	10–568	10–929	
Highest AST (U/L)				<0.001
Median (*n*)	91.0 (187)	52.0 (473)	59.0 (660)	
Min–Max	18–486	20–374	18–486	
Highest ALP (U/L)				0.049
Median (*n*)	293.0 (187)	266.0 (467)	273.5 (654)	
Min–Max	57–637	43–830	43–830	
Highest Total Bilirubin (mg/dl)				0.272
Median (*n*)	0.6 (187)	0.5 (466)	0.5 (653)	
Min–Max	0.2–16.0	0.2–16.0	0.2–16.0	
**Medication Use**				
Any Metformin Use, *n* (%)	48 (25.7%)	63 (13.3%)	111 (16.8%)	<0.001
Any Vitamin E Use, *n* (%)	67 (35.8%)	30 (6.3%)	97 (14.7%)	<0.001
Follow-up				
Time from First ALT to Last ALT (mon) ^§^				<0.001
Median (*n*)	34.9 (187)	26.2 (453)	28.5 (640)	
Min–Max	1.4–66.4	1.3–68.6	1.3–68.6	

^†^ Age calculated based on year of consent minus birth year. ^‡^ As indicated by medical history, adverse events, concomitant medications or HbA1 c > 6.5. ^§^ Where time from first ALT to time of last ALT is more than 1 month.

**Table 2 children-09-00374-t002:** Demographic and Patient Characteristics by Peak ALT Categories.

	Peak ALT			All Participants (*n* = 660)	*p*-Values
Summary	≤70 IU/L	71 to 250 IU/L	>250 IU/L		
	(*n* = 164)	(*n* = 415)	(*n* = 81)		
**Clinical Characteristics**					
Age at Study Entry (years) ^†^					Overall: 0.017
Median (*n*)	13.0 (164)	14.0 (415)	14.0 (81)	13.0 (660)	Low vs. High: 0.097
Mean (SD)	12.9 (3.04)	13.7 (2.97)	13.6 (2.62)	13.5 (2.96)	Med vs. High: 0.766
Min–Max	4–18	7–21	7–20	4–21	Low vs. Med: 0.005
Age Categories, *n* (%)					Overall: 0.119
Age < 12	53 (32.3%)	102 (24.6%)	18 (22.2%)	173 (26.2%)	Low vs. High: 0.097
Age ≥ 12	111 (67.7%)	313 (75.4%)	63 (77.8%)	487 (73.8%)	Med vs. High: 0.648
					Low vs. Med: 0.061
Gender, *n* (%)					Overall: 0.002
Female	67 (40.9%)	107 (25.8%)	24 (29.6%)	198 (30.0%)	Low vs. High: 0.084
					Med vs. High: 0.477
					Low vs. Med: 0.001
Race, *n* (%)					White vs. Non-White
White	119 (72.6%)	272 (65.5%)	50 (61.7%)	441 (66.8%)	Overall: 0.432
Black or African American	5 (3.0%)	13 (3.1%)	1 (1.2%)	19 (2.9%)	Low vs. High: 0.312
American Indian or Alaska Native	3 (1.8%)	10 (2.4%)	3 (3.7%)	16 (2.4%)	Med vs. High: 0.818
Asian	0 (0.0%)	9 (2.2%)	1 (1.2%)	10 (1.5%)	Low vs. Med: 0.231
Native Hawaiian or other Pacific Islander	1 (0.6%)	2 (0.5%)	0 (0.0%)	3 (0.5%)	
Other	27 (16.5%)	73 (17.6%)	16 (19.8%)	116 (17.6%)	
Not Reported	9 (5.5%)	36 (8.7%)	10 (12.3%)	55 (8.3%)	
Ethnicity, *n* (%)					Overall: 0.264
Hispanic or Latino	99 (60.4%)	241 (58.1%)	55 (67.9%)	395 (59.8%)	Low vs. High: 0.272
Not Hispanic or Latino	64 (39.0%)	172 (41.4%)	26 (32.1%)	262 (39.7%)	Med vs. High: 0.105
Not Reported	1 (0.6%)	2 (0.5%)	0 (0.0%)	3 (0.5%)	Low vs. Med: 0.600
Insurance Type, *n* (%)	164 (100.0%)	415 (100.0%)	81 (100.0%)	660 (100.0%)	Participants have more than one insurance type so *p*-values are not presented.
Medicaid	118 (72.0%)	285 (68.7%)	58 (71.6%)	461 (69.8%)	
Private	40 (24.4%)	109 (26.3%)	20 (24.7%)	169 (25.6%)	
Medicare	3 (1.8%)	9 (2.2%)	2 (2.5%)	14 (2.1%)	
Unknown	4 (2.4%)	9 (2.2%)	1 (1.2%)	14 (2.1%)	
Other	0 (0.0%)	4 (1.0%)	1 (1.2%)	5 (0.8%)	
Supplemental	0 (0.0%)	1 (0.2%)	1 (1.2%)	2 (0.3%)	
Uninsured	0 (0.0%)	2 (0.5%)	0 (0.0%)	2 (0.3%)	
Most Recent BMI (kg/m^2^)					Overall: 0.328
Median (*n*)	34.0 (164)	34.0 (413)	34.0 (79)	34.0 (656)	Low vs. High: 0.215 Med vs. High: 0.139 Low vs. Med: 0.897
Mean (SD)	35.0 (7.6)	34.9 (6.9)	36.2 (7.6)	35.1 (7.2)	
Min–Max	20–61	23–64	22–59	20–64	
Percentile for Most Recent BMI					Overall: 0.098
Median (*n*)	99.09 (164)	99.05 (410)	99.20 (78)	99.09 (652)	Low vs. High: 0.046
Min–Max	48.5–100.0	52.0–100.0	92.5–99.9	48.5–100.0	Med vs. High: 0.328
					Low vs. Med: 0.095
Z-Score for Most Recent BMI	2.36 (164)	2.35 (410)	2.41 (78)	2.36 (652)	Overall: 0.326
Median (*n*)	−0.0–4.5	0.1–3.3	1.4–3.1	−0.0–4.5	Low vs. High: 0.244
Min–Max					Med vs. High: 0.135
					Low vs. Med: 0.790
Cirrhosis, *n* (%)	2 (1.2%)	14 (3.4%)	8 (9.9%)	24 (3.6%)	Overall ^6^: 0.005
					Low vs. High ^6^: 0.003 Med vs. High: 0.020
					Low vs. Med ^6^: 0.123
Diabetes, *n* (%) ^‡^	27 (16.5%)	67 (16.1%)	29 (35.8%)	123 (18.6%)	Overall: <0.001
					Low vs. High: <0.001 Med vs. High: <0.001
					Low vs. Med: 0.925
**Laboratory Parameters**					
Highest ALT (U/L)					-
Median (*n*)	53.0 (164)	114.0 (415)	329.0 (81)	103.0 (660)	
Min–Max	10–70	71–250	252–929	10–929	
Highest AST (U/L)					Overall: <0.001
Median (*n*)	35.0 (164)	64.0 (415)	185.0 (81)	59.0 (660)	Low vs. High: <0.001
Min–Max	18–135	28–374	81–486	18–486	Med vs. High: <0.001
					Low vs. Med: <0.001
Highest ALP (U/L)					Overall: 0.272
Median (*n*)	274.5 (164)	266.0 (410)	308.5 (80)	273.5 (654)	Low vs. High: 0.122
Min–Max	56–541	43–830	71–620	43–830	Med vs. High: 0.143
					Low vs. Med: 0.733
Lowest Hemoglobin (g/dL)					Overall: 0.002
Median (*n*)	13.0 (161)	13.4 (397)	13.2 (78)	13.3 (636)	Low vs. High: 0.287
Min–Max	8–18	6–17	6–16	6–18	Med vs. High: 0.003
					Low vs. Med: 0.016
Highest Total Bilirubin (mg/dl)					Overall: 0.190
Median (*n*)	0.5 (163)	0.5 (410)	0.6 (80)	0.5 (653)	Low vs. High: 0.131
Min–Max	0.2–16.0	0.2–16.0	0.3–12.0	0.2–16.0	Med vs. High: 0.071
					Low vs. Med: 0.874
**Medication Use**					
Any Metformin Use, *n* (%)					Overall: 0.001
	26 (15.9%)	59 (14.2%)	26 (32.1%	111 (16.8%)	Low vs. High: 0.004
					Med vs. High: <0.001
					Low vs. Med: 0.618
Any Vitamin E Use, *n* (%)					Overall: <0.001
	10 (6.1%)	62 (14.9%)	25 (30.9%)	97 (14.7%)	Low vs. High: <0.001
					Med vs. High: 0.001
					Low vs. Med: 0.002
**Follow-up**					
Time from First ALT to Last ALT (mon) ^§^					Overall: 0.152
Median (*n*)	28.0 (155)	28.3 (405)	31.1 (80)	28.5 (640)	Low vs. High: 0.085
Min–Max	1.6–68.6	1.4–68.3	1.3–65.6	1.3–68.6	Med vs. High: 0.496
					Low vs. Med: 0.103

^†^ Age calculated based on year of consent minus birth year. ^‡^ As indicated by medical history, adverse events, concomitant medications or HbA1c > 6.5. ^§^
*p*-value is from an exact test.

**Table 3 children-09-00374-t003:** Peak ALT values by patient characteristics.

	Number of Patients (*n* = 660)	Mean (SD)	Median (Min–Max)	*p*-Value
Age at enrollment (yrs)				
Age < 12	173	133.3 (122.3)	98.0 (22–819)	0.44
Age ≥ 12	487	141.1 (109.4)	104.0 (10–929)	
Sex				0.186
Female	198	130.2 (110.8)	92.0 (10–694)	
Male	462	142.8 (113.7)	106.5 (14–929)	
Race				0.327 ^†^
White	441	133.1 (108.6)	98.0 (14–929)	
Black or African American	19	94.9 (58.5)	78.0 (29–257)	
American Indian or Alaska Native	16	160.8 (97.1)	125.0 (46–360)	
Asian	10	145.8 (101.8)	98.0 (73–383)	
Native Hawaiian or other Pacific Islander	3	74.0 (28.1)	76.0 (45–101)	
Other Race	116	149.4 (123.1)	112.0 (10–715)	
Ethnicity				0.018
Hispanic or Latino	395	147.7 (123.3)	106.0 (19–929)	
Not Hispanic or Latino	262	126.4 (94.5)	98.5 (10–819)	
Body Mass Index, Percentile ^‡^				0.167
≤95 th	47	116.8 (71.05)	105.0 (10–357)	
>95 th	605	140.4 (115.2)	103.0 (14–929)	
Cirrhosis				0.018
Yes	24	192.8 (125.0)	155.5 (27–473)	
No	636	137.0 (120.0)	102.0 (10–929)	
History of Type 2 diabetes				0.004
Yes	123	171.2 (121.8)	137.0 (14–615)	
No	537	131.7 (109.5)	99.0 (10–929)	
Any Metformin Use				0.003
Yes	111	167.7 (122.6)	130.0 (14–615)	
No	549	133.2 (110.0)	100.0 (10–929)	
Any Vitamin E Use				<0.001
Yes	97	204.7 (158.7)	162.0 (26–929)	
No	563	127.7 (98.9)	98.0 (10–715)	
Biopsy				<0.001
Yes	187	211.2 (150.7)	169.0 (28–929)	
No	473	110.5 (77.21)	88.0 (10–568)	

*p*-values comparing mean peak ALT. ^†^*p*-Value comparing white vs. non-white. ^‡^ BMI Percentile is calculated from most recent height and weight measurements closest to or at enrollment. Any medication use and history of co-morbid disease were ascertained prior to or at enrollment. Weight percentile was calculated based on the most recent height and weight values closest to enrollment using CDC 2000 growth charts.

## Data Availability

Due to privacy issues, data is not publicly available.

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
