# Peer review of "Variation in Alanine Aminotransferase in Children with Non-Alcoholic Fatty Liver Disease"

_children, 2022, doi:10.3390/children9030374_

Round 1
Reviewer 1 Report
This study published by Castillo-Leon et al. based on data from TARGET-NASH observational cohort describes the variation of ALT in NALFD patients and the correlation with different disease characteristics. It is a real-world longitudinal observational study, including a large number of children with NAFLD.
The paper is well written, presenting interesting and important data, but there are some improvements regarding the presentation of the results, discussions and editing.
- Keywords should be improved. Why cancer?
- Table 1: at cirrhosis, I recommend leaving just data for the presence of cirrhosis (as the other are clear the difference); for diabetes, just the data (no need for yes; it is obvious); same for any metformin use and any vitamin E use. The same recommendation for Table 2
- Table 3: some horizontal lines to separate the different categories
- Figure 1: verify if the data presented are the mean peak ALT, in correlation with Table 3; there should be a title for the figure
- Figure 2 - there is no need for "Figure 2 legend" text, but the Figure should have a title
- The discussion should be improved with correlations with results from other studies. This section is short compared to the other sections.
- There are no conclusions presented in the paper, and a section should be added at the end of the manuscript.
- Editing: the abbreviated words should be used consistently after their first definition: see lines 45, 115, 208, 274
Author Response
The paper is well written, presenting interesting and important data, but there are some improvements regarding the presentation of the results, discussions and editing.
Keywords should be improved.
Thank you for this feedback. We will work with the editing team to update the keyword selection to something more appropriate.
Table 1: at cirrhosis, I recommend leaving just data for the presence of cirrhosis (as the other are clear the difference); for diabetes, just the data (no need for yes; it is obvious); same for any metformin use and any vitamin E use. The same recommendation for Table 2
Revisions have been made to Table 1 and Table 2 to reflect these recommended changes.
Table 3: some horizontal lines to separate the different categories
Thank you for this feedback. Horizontal lines have been added to Table 3.
Figure 1: verify if the data presented are the mean peak ALT, in correlation with Table 3; there should be a title for the figure
The data in figure 1 depicts the median and quartile range of Peak ALT. The title of figure 3 has been updated accordingly to be more descriptive.
Figure 2 - there is no need for "Figure 2 legend" text, but the Figure should have a title
Thank you for your feedback. A figure title has been inserted for Figure 2 and the Figure 2 legend has been removed.
The discussion should be improved with correlations with results from other studies. This section is short compared to the other sections.
The discussion section has been updated to incorporate more references/correlations to results from other studies.
There are no conclusions presented in the paper, and a section should be added at the end of the manuscript.
Thank you for pointing this out. Conclusions have been added to the end of the manuscript.
Editing: the abbreviated words should be used consistently after their first definition: see lines 45, 115, 208, 274
Thank you. The manuscript was revised to ensure that abbreviated words were used consistently.
Reviewer 2 Report
This is an interesting topic, but some clarification should be added.
- Did you get the approval of the Ethics Committee from the hospitals / academic institution before starting the study? Please add the approval number in the article.
- Did the children have any other comorbidities other than diabetes and obesity?
- Why did children take metformin? Usually, in children, diabetes is type 1 and it is treated with insulin.
- Did the children use drugs with liver toxicity during the study (eg. acetaminophen)?
- After being diagnosed with NAFLD, were the children treated with hepatoprotective agents?
- Did cirrhosis have a specific cause (eg. viral infection)?
- Why was vitamin E prescribed for these children?
- The conclusion section was omitted from the article.
Author Response
This is an interesting topic, but some clarification should be added.
- Did you get the approval of the Ethics Committee from the hospitals / academic institution before starting the study? Please add the approval number in the article.
Yes, approval was obtained from the Western Institutional Review Board, approval code WCG 2016138. This information has been added to the manuscript.
- Did the children have any other comorbidities other than diabetes and obesity?
Yes, the children in our study had other comorbidities as well. Of the 660 participants, 101 patients (15.3%) had a history of anxiety or depression, 89 patients (13.5%) had a history of pediatric ADHD, and 71 patients (10.8%) had a history of psoriasis or other autoimmune skin disease.
- Why did children take metformin? Usually, in children, diabetes is type 1 and it is treated with insulin.
- Did the children use drugs with liver toxicity during the study (eg. acetaminophen)?
These patients have been evaluated at hepatology centers by expert pediatric hepatologists, and hepatotoxicity was not considered a likely etiology for elevated ALT in this population. After further exploration into the medications prescribed to patients within our sample, greater than 10% of participants used the following medications: analgesics, corticosteroids, obstructive airway disease, vitamins, etc. However, there were only prescribing differences found across peak ALT categories for diabetes medications (p=0.0003) and tonics (Tocopherol/Vitamin E) (p<0.0001), which were deemed unlikely to be responsible for variations in ALT.
- After being diagnosed with NAFLD, were the children treated with hepatoprotective agents?
Hepatoprotective drugs for pediatric patients with NAFLD is limited to Vitamin E, which has shown some improvement in ALT (3). At present, there are no NAFLD medications for use in children (4).
- Did cirrhosis have a specific cause (eg. viral infection)?
Patients in our study did not have any indication of viral hepatitis.
- Why was vitamin E prescribed for these children?
As discussed in comment 5, there are no medications for use in children with NAFLD. At present, Vitamin E is the only treatment that has been shown to have some impact on ALT levels (3).
- The conclusion section was omitted from the article.
Thank you for pointing this out. Conclusions have been added to the end of the manuscript.
- Association AD. Type 2 diabetes in children and adolescents. Pediatrics. 2000;105(3):671-80.
- Pettitt DJ, Talton J, Dabelea D, Divers J, Imperatore G, Lawrence JM, et al. Prevalence of diabetes in US youth in 2009: the SEARCH for diabetes in youth study. Diabetes care. 2014;37(2):402-8.
- Lavine JE, Schwimmer JB, Van Natta ML, Molleston JP, Murray KF, Rosenthal P, et al. Effect of vitamin E or metformin for treatment of nonalcoholic fatty liver disease in children and adolescents: the TONIC randomized controlled trial. Jama. 2011;305(16):1659-68.
- Crudele A, Panera N, Braghini MR, Balsano C, Alisi A. The pharmacological treatment of nonalcoholic fatty liver disease in children. Expert Review of Clinical Pharmacology. 2020;13(11):1219-27.
Round 2
Reviewer 1 Report
The authors improved the manuscript following the recommendations. Still, for the keywords, I do not understand why "cancer" is there and no transaminases or other words relevant for this study.
Author Response
Thank you for this feedback. The keyword "cancer" has been replaced with ALT.